

# Analysis of copper, selenium, and zinc in newborn dried bloodspots using total reflection X-ray fluorescence (TXRF) spectroscopy

Jessica Pawly[1], Richard L. Neitzel[2] and Niladri Basu[1,2]

[1] Faculty of Agricultural and Environmental Sciences, McGill University, Montreal, Canada
[2] University of Michigan, Ann Arbor, United States of America

## ABSTRACT

**Background**. There exists great interest in using dried bloodspots across the clinical, public health, and nutritional sciences to characterize circulating levels of essential elements yet current methods face several challenges related to instrumentation, quality control, and matrix effects. Elemental analysis via total X-ray fluorescence (TXRF) may help overcome these challenges. The objective of this study was to develop and apply a novel TXRF-based analytical method to quantify essential elements (copper, selenium, zinc) in dried bloodspots.

**Methods**. Analytical methods were developed with human whole blood standard reference materials from the Institut National de Santé Publique du Québec (INSPQ). The method was developed in careful consideration of several quality control parameters (e.g., analytical accuracy, precision, linearity, and assay range) which were iteratively investigated to help refine and realize a robust method. The developed method was then applied to a quantitative descriptive survey of punches ($n = 675$) taken from residual dried bloodspots from a newborn screening biobank program (Michigan BioTrust for Health).

**Results**. The analytical method developed to quantify the three target elements in dried bloodspots fared well against a priori quality control criteria (i.e., analytical accuracy, precision, linearity and range). In applying this new method, the average ($\pm$SD) blood copper, selenium, and zinc levels in the newborn samples were 1,117.0 $\pm$ 627.1 µg/L, 193.1 $\pm$ 49.1 µg/L, and 4,485 $\pm$ 2,275 µg/L respectively. All the elements were normally distributed in the sample population, and the measured concentrations fall within an expected range.

**Conclusions**. This study developed and applied a novel and robust method to simultaneously quantify three essential elements. The method helps overcome challenges in the field concerning elemental analysis in dried bloodspots and the findings help increase understanding of nutritional status in newborns.

Corresponding author
Niladri Basu, niladri.basu@mcgill.ca

## INTRODUCTION

Trace element quantification is critical in activities such as clinical assessments, nutritional research, and public health interventions (*Institute of Medicine, 2000*; *Institute of Medicine, 2001*). There remains increasing interest in the development and use of biomarkers to study such elements (*Potischman, 2003*; *Mcdade, Williams & Snodgrass, 2007*) though many biomarkers remain costly to analyze and suffer from logistical and ethical constraints. For example, venous whole blood is frequently viewed as the 'gold standard' for many analytes though its collection is invasive and sampling requires trained professionals and specialized supplies.

Dried blood spots (DBS) are being hailed as an alternative to venipuncture (*Mcdade, Williams & Snodgrass, 2007*; *Mei et al., 2001*). They provide a minimally invasive and low-cost method for collecting capillary blood onto filter paper, and thus represent an ethical, practical, and economical alternative to venipuncture. Among other benefits (*Mcdade, Williams & Snodgrass, 2007*; *Mei et al., 2001*), the blood does not need to be processed after collection thus reducing complexities associated with sampling, transport, and storage (*Boerma, Holt & Black, 2001*; *Garrett et al., 2001*). Despite the potential benefits of DBS for elemental analysis, outstanding gaps have impaired widespread adoption (*Langer, Johnson & Shafer, 2010*; *Vacchina et al., 2014*; *Marguí et al., 2017*; *Basu et al., 2017*; *Perkins & Basu, 2018*). Notably, quantification of target essential elements in DBS often require instrumentation that can achieve low detection limits while maintaining accuracy and precision, and the effects of storage temperature and method as well as the paper filter matrix need careful consideration.

Moving forward, some of the aforementioned challenges associated with the measurement of essential elements in DBS may be overcome through the use of Total Reflection X-ray Fluorescence (TXRF) Spectroscopy. Comparisons with accepted methods of elemental analysis such as atomic absorption spectroscopy (AAS) and inductively-coupled plasma mass spectroscopy (ICPMS) demonstrate that TXRF is a practical, accurate, and reliable alternative (*Borgese et al., 2009*; *Stosnach, 2005*; *Stosnach & Mages, 2009*). Like these spectroscopy methods, TXRF can detect multiple elements in a range of sample types though for TXRF the approach entails simpler preparation methods and reduced sample volumes and run times, and these in turn help reduce analytical costs. In addition, the matrix effect is minimized in TXRF as aqueous samples are dried onto a quartz carrier disc thus reducing absorption or secondary excitation (*Towett et al., 2015*). This may be particularly advantageous for studying elements in DBS.

The objective of this study was to develop and apply a novel method of quantifying essential elements in DBS using TXRF spectroscopy. The elements selected for this study were copper (Cu), selenium (Se), and zinc (Zn). We focused on these elements as they are classified as "essential" elements and more specifically they play essential roles in metabolism, have antioxidant properties, and mediate proper reproductive and other important health outcomes as reviewed by the US Institute of Medicine (IOM) Panel on Micronutrients (*Institute of Medicine, 2000*; *Institute of Medicine, 2001*). For each of these elements we first developed a new analytical assay and then we applied the developed

method to measure these three elements in residual DBS obtained from newborns ($n = 675$) from the State of Michigan's newborn screening biobank program (Michigan BioTrust for Health).

## MATERIALS & METHODS

### General overview

In study phase #1, the method was developed using human whole blood standard reference material (SRM) with assigned elemental concentrations (Table S1). Artificial DBS were created in the laboratory using these SRMs, and then these DBS were used to evaluate several quality control criteria (i.e., assay linearity, range, accuracy, precision) (*Li & Lee, 2014*) to help establish a suitable analytical method. Next, in study phase #2, the established method was applied to quantify essential elements in DBS from newborns ($n = 675$) from the Michigan BioTrust for Health program as part of a larger effort concerning newborn hearing loss. The elements focused on were copper (Cu), selenium (Se), and zinc (Zn). While TXRF can detect multiple elements, we focused the current work to develop methods for these three essential elements which play key roles in nutrition, health and disease (*Institute of Medicine, 2000*; *Institute of Medicine, 2001*).

Institutional Review Board (IRB) approval for this work was obtained from McGill University (A06-M29-16B), the University of Michigan (HUM000771006), and the Michigan Department of Health and Human Services (201212-05-XA-R).

### Dried bloodspots

In the methods development phase, human whole blood SRM ($n = 7$) of varying elemental concentrations from the Institut National de Santé Publique du Québec (INSPQ) were used (Table S1). Artificial DBS were created by pipetting a 60 µL sample of whole blood SRM onto Whatman©903 filter paper (GEO Healthcare Services, Mississauga, ON, Canada) and dried overnight at room temperature in a Class 100 ISO Cleanhood. After drying, the DBS cards were stored in plastic bags at ambient temperature until use. The 60 µL DBS was sub-sampled using a three mm punch (Harris Corporation, Melbourne, FL, USA). A punch of this size is often used in studies of DBS and assumed to contain 3.1 µL of blood (*Li & Lee, 2014*). Blank filter paper adjacent to the DBS was also analyzed from approximately 10% of all DBS cards. Punched DBS samples were placed in a metal-free microcentrifuge tube (Rose Scientific Ltd.) until analysis. We then used these punched DBS samples to evaluate a series of parameters (Table S2) to yield a suitable analytical method in terms of analytical accuracy, precision, linearity and range as per bioanalytical assay development recommendations from ICH (Reports Q2A and Q2B) and ISO 17025 as summarized in *Huber (2007)*.

In study phase #2, we focused on newborn DBS collected between 2003 and 2015 from the Michigan BioTrust for Health. The analyses of these samples spanned 38 batch runs with each batch containing a maximum of 24 samples composed of 18 individual newborn DBS samples, 3 DBS SRMs, 2 method blanks, and 1 DBS sample from which a duplicate punch was analyzed. For batch runs #1-10 ($n = 180$), the DBS were stored at ambient temperatures until analysis. The rest of the samples ($n = 495$) were stored at $-20$ °C

between collection and analyses. For the first 12 batches analyzed, one punch (three mm diameter) was taken from the edge of a single spot of a DBS card. For the remaining batches, the Michigan BioTrust for Health provided rectangular punches of two mm × six mm in size, or the equivalent of two three mm diameter punches.

To bring the dried blood into solution, 15 µL of concentrated HCl with five mmol EDTA was added to the three mm punch in the microcentrifuge tube. Note, in the case of the rectangular punches, double volumes were applied. The sample was then vortexed thoroughly and digested for 1.5 h at 55 °C. Following the digestion, the sample was centrifuged for 15 min at 25 °C at 12,000 rpm. An 8 µL portion of the extraction fluid was removed and placed into a second microcentrifuge vial, to which a 4 µL solution containing a mixture of gallium (internal standard, 100 µg/L final concentration) and polyvinyl alcohol (1% vol/vol) was added. This solution was mixed, and then an 8 µL aliquot was placed onto a Serva-conditioned quartz sample disc carrier. The sample was covered and allowed to dry overnight in a lab oven set at 55 °C following which analysis was performed. A representative photo of an extract dried onto the sample disc carrier is provided (Fig. S1).

## Multi-element analysis

Multi-element measurement (Cu, Se, Zn) was carried out using TXRF spectroscopy (S2 PICOFOX, Bruker AXS Microanalysis GmbH, Germany; technical specifications are in Table S3) as detailed previously by others (*Stosnach & Mages, 2009*). Samples were read for 2,500 s and the results were analyzed using the instrument's software, *Spectra 7* (Bruker AXS Inc.). Elemental concentration was quantified using a gallium internal standard and a seven-point matrix-matched calibration curve for each element. A representative spectra is provided (Fig. S2).

## Quality control

Each run batch contained a maximum of 24 samples including a range of quality control samples as previously detailed. The punched DBS made from SRMs ($n = 7$; Table S1) were used to establish matrix-matched calibration curves and characterize analytical accuracy, precision, linearity, and assay range. Analytical percent accuracy was calculated as the difference between the observed value and the accepted concentration value of the SRM. Intra-day assay precision was assessed by analyzing DBS samples from the Michigan BioTrust for Health study in duplicate, and calculated as the relative percent difference between the two measures. Inter-day assay precision was assessed by comparing the values of the SRMs across batch runs with a coefficient of variability (%CV) calculated. In order to determine background levels of elements of the filter paper, we analyzed blank filter paper removed from adjacent areas to the spots of blood of select DBS samples. The elemental concentrations of the filter paper were not subtracted from the results of the accompanying blood spot as discussed further below.

## Data analyses

Data from *Spectra 7* (Bruker AXS Inc.) was first analyzed using descriptive statistics and graphical plots to understand basic features of the dataset. For the methods development phase of the work, findings were compared against assay performance criteria (i.e., assay linearity, range, accuracy, precision) (*Huber, 2007*) to help establish a working method. For the application phase of this work (i.e., analysis of newborn DBS from the Michigan Biotrust for Health), measures of central tendency (mean, median) and associated variances (standard deviation, inter-quartile ranges) were calculated. Further, $t$-tests and ANOVAs were run to test if elemental levels varied according to batch number ($n = 38$), punch type (three mm diameter vs. rectangular punch), and storage temperature (ambient vs. $-20\,°C$). Outliers were detected using the generalized extreme studentized deviate (ESD) test. The $p$-value was set at $\alpha = 0.05$ for all tests. All analyses were conducted using Microsoft® Excel Version 15.4 and JMP®Pro 13.0.0. Data are represented as mean $\pm$ standard deviation unless otherwise noted.

## RESULTS

### Assay quality control

The linearity of the developed method was assessed by measuring elemental concentrations in DBS created in the laboratory with seven different whole blood SRMs (purposefully chosen as their assigned values spanned a range of concentrations deemed to be physiologically relevant). From these artificially created DBS, sub-samples were punched from the edge of a DBS and were analyzed individually. The average of three samples was calculated and compared to the assigned elemental concentration to develop a matrix-matched calibration curve, and to also assess linearity and accuracy.

For Cu, the resulting linear regression comparing concentrations analyzed on the TXRF (i.e., punches of DBS taken from whole blood SRM added to filter paper cards) and the assigned SRM concentrations in the whole blood was $Y = 0.99X - 7.2$, with a coefficient of determination ($R^2$) of 0.98 (Fig. S3). The assay linearity for Cu was thus deemed to be acceptable. The average recovery of Cu from the DBS of the SRMs during the methods development phase was $102.3 \pm 5.9\%$, and later during the application phase ranged from 100.3 to 117.2% for the 3 focal SRMs used (Table 1). For Se, the resulting linear regression comparing concentrations analyzed on the TXRF (of the DBS) and the assigned SRM concentrations in the whole blood was $Y = 1.2X + 9.6$, with a $R^2$ of 0.98 (Fig. S4). The assay linearity was thus deemed to be acceptable. The average recovery of Se from the DBS of the SRMs during the methods development phase was $100.9\% \pm 8.6$ and later during the application phase ranged from 90.9 to 95.7% for the 3 focal SRMs used (Table 1). For Zn, the resulting linear regression comparing concentrations analyzed on the TXRF (of the DBS) and the assigned SRM concentrations in the whole blood was $Y = 1.1X - 585.5$, with a coefficient of determination of 0.975 (Fig. S5). As with Se and Cu, the assay linearity was again deemed to be acceptable. The average recovery of Zn from the DBS of the SRMs during the methods development phase was $102.3 \pm 5.6\%$, and later during the application phase ranged from 104.9 to 115.2% for the 3 focal SRMs used (Table 1).

**Table 1** Analytical accuracy and precision of elemental measurements taken in dried blood spots using different whole blood standard reference materials.

| Element | SRM | Assigned SRM Concentration (µg/L) | Measured Concentration in DBS (µg/L) | Accuracy (%) | Precision (% CV) |
|---|---|---|---|---|---|
| Copper (Cu) | QM-B-Q1313 | 3,094.9 | 3,103.7 ± 365.9 | 100.3 ± 11.8 | 11.8 |
| | QM-B-Q1505 | 813.4 | 953.4 ± 160.2 | 117.2 ± 19.6 | 16.8 |
| | QM-B-Q1506 | 3,037.7 | 3,215.4 ± 349.3 | 105.8 ± 11.5 | 10.9 |
| Selenium (Se) | QM-B-Q1313 | 290.6 | 264.4 ± 35.9 | 90.9 ± 12.2 | 13.6 |
| | QM-B-Q1505 | 172.1 | 158.2 ± 36.7 | 91.9 ± 21.3 | 23.2 |
| | QM-B-Q1506 | 226.6 | 216.9 ± 33.1 | 95.7 ± 14.6 | 15.3 |
| Zinc (Zn) | QM-B-Q1313 | 7,910.9 | 8,297.1 ± 1,104.1 | 104.9 ± 13.9 | 13.3 |
| | QM-B-Q1505 | 6,335.3 | 7,297.5 ± 1,145.7 | 115.2 ± 18.1 | 15.7 |
| | QM-B-Q1506 | 10,853.1 | 11,591.1 ± 1,412.1 | 106.8 ± 13.0 | 12.2 |

Assay precision was evaluated using two different approaches. First, inter-assay precision (as % CV) was evaluated by comparing the SRMs analyzed across the run batches. SRMs from Batch 1 were damaged prior to analysis due to human error and the internal standard of SRMs from Batch 30 were not quantified, so those values were removed. Second, intra-assay precision (expressed as % relative percent difference, RPD) was addressed by measuring two individual punches from a given sample in the Michigan BioTrust for Health samples. The inter- and intra-assay precision of the three selected SRMs for Cu, Se and Zn are provided (Table 1). For Cu, the inter-assay precision of the SRM samples ranged from 10.9 to 16.8%, and the intra-assay precision ranged from 0.8 to 39.6%; five of the replicates had a %RPD greater than 30% (Fig. S6). For Se, the inter-assay precision of the SRM samples ranged from 13.6 to 23.2%, and the intra-assay precision ranged from 0.1 to 53.2%; three of the replicates had a % RPD greater than 30% (Fig. S7). For Zn, the inter-assay precision ranged from 12.2 to 15.7%, and the intra-assay precision ranged from 0.4% to 70.5%; four of the replicates had a %RPD greater than 30% (Fig. S8).

The elements were measured in punches of blank filter paper taken adjacent to the DBS. One three mm circular punch of blank filter paper was taken from approximately 10% of the blotted DBS cards analyzed in the Michigan BioTrust for Health cohort and two blank filter papers were analyzed in every batch. For Cu, the average of all the blank filter papers was 818.0 ± 1413.9 µg/L. However, this value is driven by two extremely high blank values (6,194.8 and 8,262.4 µg/L). When these values are removed the resulting average was 629.5 ± 874.1 µg/L. Se was only detected in one sample of blank filter paper at a concentration of 13.4 µg/L. For Zn, the average of all the blank filter paper measurements was 2,005.3 ± 1,622.2 µg/L.

## Assay application: Michigan BioTrust for Health Project

The three target essential elements were measured in DBS samples obtained from 675 newborns from the Michigan BioTrust for Health Project. For each element some outliers

**Table 2  Concentrations (μg/L) of copper, selenium and zinc measured in newborn dried bloodspot samples.**

| Element | Sample size | Mean | SD | Percentile | | | | | | |
|---------|-------------|------|----|-----|------|------|------|------|------|------|
| | | | | 5 | 10 | 25 | 50 | 75 | 90 | 95 |
| Copper | 673 | 1,117.0 | 627.1 | 669.8 | 735.0 | 844.9 | 997.8 | 1,202.6 | 1,498.4 | 1,947.9 |
| Selenium | 660 | 193.1 | 49.1 | 118.7 | 134.2 | 161.0 | 188.8 | 225.1 | 256.3 | 273.1 |
| Zinc | 673 | 4,485.1 | 2,274.9 | 2,439.3 | 2,748.3 | 3,382.0 | 4,069.6 | 4,970.2 | 6,301.0 | 7,359.1 |

were identified through the generalized ESD test and the final sample size for each element is provided in Table 2.

The average (SD) blood Cu, Se, and Zn levels in the punches sampled from the newborn DBS were 1,117.0 ± 627.1 μg/L, 193.1 ± 49.1 μg/L, and 4,485 ± 2,275 μg/L respectively (Table 2; Fig. 1). All the elements were normally distributed in the sample population (Fig. S9).

In terms of variation across the 38 different batch runs, some differences were calculated for each of the 3 elements with mean values in some of the batches being significantly different (Fig. 1). Specifically, for Cu, (3 batches), Se (5 batches), and Zn (7 batches), the number of batches that were significantly different are indicated in brackets. Variation within and across batches is expected, and we did not find any systemic bias as other quality control measures across the batches (e.g., inter-assay precision measures) performed well.

We also investigated the potential variation between the two different types of punches. The first 200 samples were punched using a three mm circular punch from the perimeter of the DBS in the lab while the remaining samples were provided by the Michigan BioTrust for Health as rectangular punches of two mm ×six mm in size. Therefore, the location of the latter samples on the DBS is unknown and could be from the center or edge, increasing the potential variability due to differences in rates of deposition of elements across the DBS. The mean results of the two different punches were compared and there were no statistically significant differences for any of the elements.

The newborn DBS samples were collected from 2003 to 2015 and were analyzed in the lab in 2016 thus indicating a maximum potential storage time of 13 years. Some of the samples were stored at room temperature whereas others were stored frozen. The DBS cards that were stored at an ambient temperature ($n = 380$) prior to analysis versus the 295 cards that were kept frozen at $-20$ °C had Cu and Se measurements that were not different. For Zn, values in the frozen cards were significantly ($p < 0.001$) lower than those taken from the cards stored at ambient temperature (3,959.6 versus 4,892.7 ug/L).

## DISCUSSION

This study developed and applied a novel method to simultaneously quantify 3 essential elements in DBS of relevance to clinical and public health sciences. While DBS are being hailed as a cost-effective, minimally invasive, and thus practical and attractive method for collecting blood from clinical and other settings (e.g., remote field sites) (*Mcdade, Williams & Snodgrass, 2007*) there remain outstanding questions particularly related to the quality of

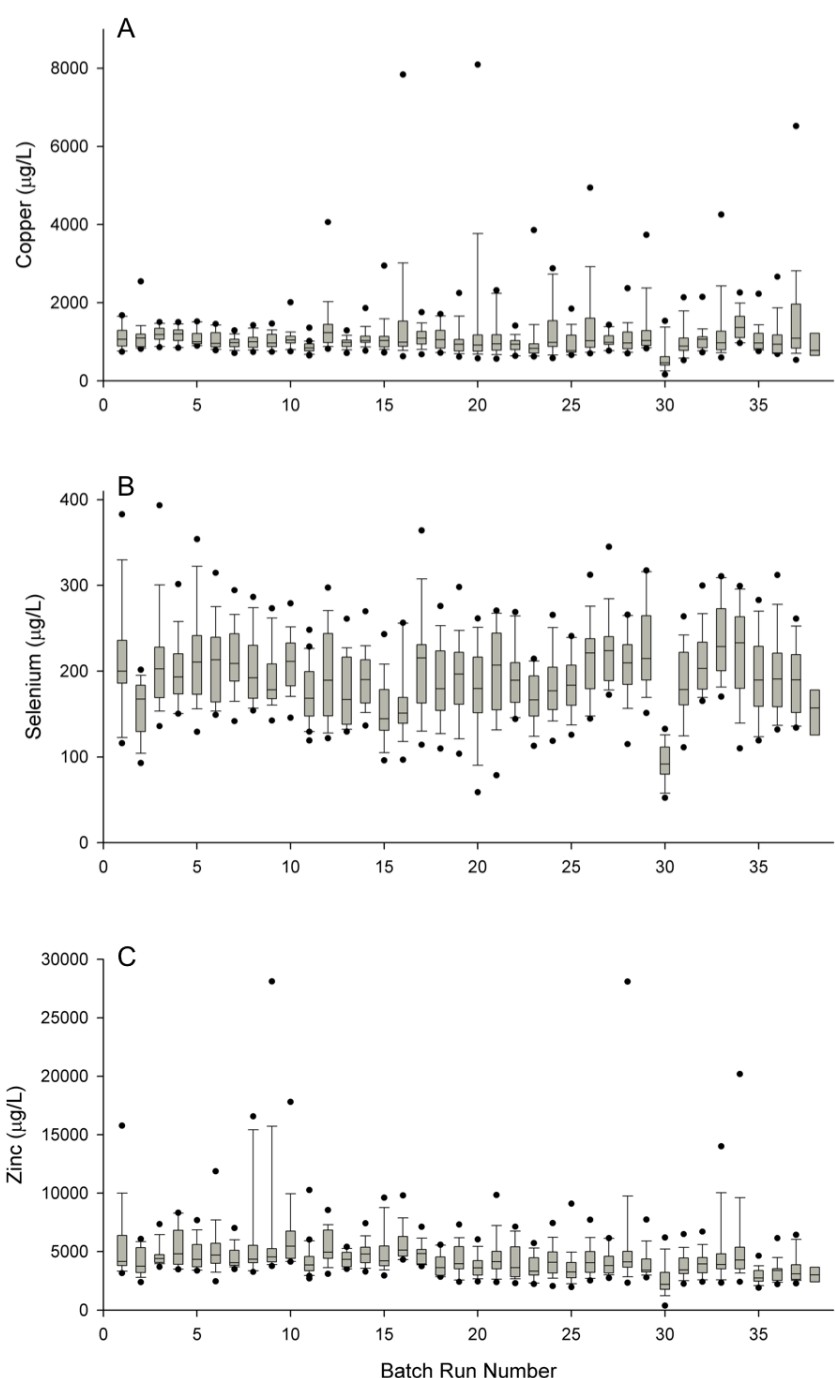

**Figure 1** Boxplots of copper (A), selenium (B), and zinc (C) measurements taken in newborn dried blood spots from the Michigan BioTrust for Health program according to batch runs.

the analytical measurements taken (*Mei et al., 2001*; *Huber, 2007*). The work presented here helps overcome some challenges present in the literature concerning elemental analysis in DBS.

The current study established a working method by carefully examining relevant quality control parameters, and then applied this method to a large cohort of 675 newborns in order to increase understanding of blood levels of Cu, Se, and Zn in newborns. The importance of these target elements, especially in newborns, is firmly established (*Institute of Medicine, 2000*; *Institute of Medicine, 2001*) though to generalize the results from the current work to a broader population has been challenging as few comparison populations exist; few studies have quantified essential elements in early lifestage groups owing to aforementioned ethical and practical challenges associated with sampling venous blood particularly from newborns (as well as infants). Thus, the development of an analytical method using DBS opens up possibilities especially since these samples are collected from newborns as part of screening programs that are routine, and in some cases required by law, in many jurisdictions (*Mei et al., 2001*; *Huber, 2007*; *Olshan, 2007*).

The average blood Cu in the newborns studied here was 1,117.0 ± 627.1 µg/L, and this is comparable to a reference value of blood Cu in adults (970 ± 130 µg/L) (*Iyengar & Woittiez, 1988*). We are not aware of another review reporting upon Cu in newborn blood though *Krachler, Rossipal & Micetic-Turk (1999)* measured this element in serum and reported a range of values between 590 and 1,390 µg/L. Further, in a study of 3,210 children aged 0-14 from Lu'an (China) the median whole blood copper level in a subset of 397 children below the age of 1 was about 1,306 µg/L with a range of 758–2,420 µg/L (*Zhai et al., 2017*). For blood Se, the concentrations measured here in the newborns (193.1 ± 49.1 µg/L) were at the upper end of the adult reference range (58 to 234 µg/L) reported on a review of 20 datasets (*Iyengar & Woittiez, 1988*). We are not aware of another dataset reporting upon Se in newborn whole blood though *Galinier et al. (2005)* established a reference range of 47.4 ± 7.9 µg/L in the umbilical cord serum of neonates ($n = 241$). For blood Zn, the average concentrations measured here in the newborn DBS was 4,485 ± 2,274 µg/L. In the aforementioned study from Lu'an, China, the median whole blood zinc level in a subset of 397 children below the age of 1 was about 4,315 µg/L with a min-max range of 3,360–6,479 µg/L (*Zhai et al., 2017*). In adults, for comparison, *Iyengar & Woittiez (1988)* determined blood Zn concentrations to be 6,500 ± 1,100 µg/L. In general, these comparisons suggest that the data from the current study are within an expected range though they also emphasize the need for more information on this topic so that adequate reference ranges can be established for newborns.

The measurements taken in the current study were carefully evaluated for several quality control criteria by consulting the resource of Huber (*Li & Lee, 2014*), and in general the developed method performed well in terms of linearity, range, accuracy and precision. Nonetheless, there are always areas to improve upon. For example, here the internal gallium standard was added to the processed sample and in the future this standard could be added to the DBS prior to processing. With respect to the presence of these elements in the blank filter paper, there were some potential challenges noted with the Cu and Zn data. Contamination in the filter paper may arise during the manufacturing of the cards,

blood collection, storage, transportation as well as sample preparation. Several studies have found high variability in elemental levels within a card and across cards (*Langer, Johnson & Shafer, 2010*; *Marguí et al., 2017*; *Chaudhuri et al., 2009*; *Hsieh et al., 2011*). The variability in contamination could explain some of the outliers in this analysis and thus serve as a barrier to accurate elemental quantification of DBS. Unfortunately, here we could not account for such variation on an individual sample basis though in future studies one may consider analyzing paired sample punches (i.e., DBS punch and a nearby punch of the blank filter paper) from each card.

The work presented here helps overcome some challenges present in the literature concerning elemental analysis though there are notable study limitations that warrant attention. One of the greatest challenges in the analysis of DBS is the unknown blood volume in a punch. While punches allow for a consistent area to be analyzed the sample volume remains unknown due to variances in sample collection methods and human physiology. For example, an individual's hematocrit affects the spread of blood on the filter paper, and this affects the blood volume in a given punch. Hematocrit has a wide range of normal values that can change based on sex, age, and health status (*O'Mara et al., 2011*). Furthermore, the hematocrit range for infants under two years of age is from 28% to 55%, compared to 41% to 50% in adult males and 36% to 44% in adult females (*Wong & James, 2014*). As a result, it is difficult to calculate an accurate concentration and here we adopted an estimation that a three mm diameter punch contains a 3.1 µL volume (*Huber, 2007*). Researchers have also shown that there may be differences in analyte distribution between the perimeter and center of the blood spot (*Perkins & Basu, 2018*; *O'Mara et al., 2011*).

Normalization may help overcome contamination in elemental analysis as well as account for unknown sample volume. Normalization can be done with one or multiple elements that have a narrow physiological distribution and absence from the blank filter paper (*Langer, Johnson & Shafer, 2010*; *Stove et al., 2012*). *Langer, Johnson & Shafer (2010)* suggested the use of potassium (K) as it is found at low levels in unspotted filter paper and could be used to normalize volume differences. Additional elements, such as magnesium (Mg) and calcium (Ca), have a narrow range in blood and could also be used to normalize values. However, *Langer, Johnson & Shafer (2010)* suggests they may be inappropriate due to high concentrations and variability in blank filter paper samples. Although these elements can be quantified with the current TXRF method, the SRMs used in this study are not certified for Ca, K, or Mg and, thus, accuracy and precision currently cannot be calculated to explore this method. We do note that the SRMs used in the current study have been assigned values for other notable elements both toxic (e.g., Pb, Cd) and essential (e.g., Cr, Mn). Moving ahead, the multi-element capabilities of TXRF make it a promising instrument to concurrently measure a range of elements and thus future work is necessary (e.g., develop and validate methods for other elements; generate appropriate reference materials with assigned levels of other elements) to enable such expansion particularly in this era of the exposome.

## CONCLUSIONS

The objective of this study was to develop and apply a novel TXRF-based analytical method to quantify essential elements (copper, selenium, zinc) in DBS. While there is great demand across the clinical, public health and ecological sciences for such a method, current approaches face several challenges related to instrumentation, quality control, and matrix effects. Here we demonstrate that elemental analysis of DBS with TXRF may help overcome these challenges, and that the developed method can be scaled-up in relatively large study settings. The analytical method developed to quantify the 3 target elements fared well against a priori quality control criteria (i.e., analytical accuracy, precision, linearity and range). We demonstrate the possibility of using the method to characterize essential elements in DBS from residual newborn screening programs, and by extension the method can be extended to a range of other settings such as, for example, demographic surveys in low- and middle-income countries and research in remote sites where there remain logistical barriers to sampling venous whole blood.

## ACKNOWLEDGEMENTS

We thank the Michigan BioTrust for Health and Michigan Department of Health program staff for their support, with particular thanks to Carrie Langbo and Nancy Christ. We also thank Jenny Eng, Krystin Carlson, Marie Perkins, Gordana Martincevec, Lisa Bidinosti, Mark Bradley, Andrea Santa Rios, Stephane Bayen, Helene Lalande, Hugo Melgar-Quinonez, and Jonathan Chevrier for their assistance with this study.

### Funding

This work was supported by the Gerber Foundation, the Michigan Bloodspot Environmental Epidemiology Project (BLEEP), Frederick Banting and Charles Best Canada Graduate Scholarship from the Canadian Institute of Health Research, Fonds de Recherche du Québec en Santé, the Canada Research Chairs (CRC) program, McGill University, infrastructure support from Canada Foundation for Innovation, and a Natural Sciences and Engineering Research Council of Canada (NSERC) Discovery Grant. The funders had no role in study design, data collection and analysis, decision to publish, or preparation of the manuscript.

### Grant Disclosures

The following grant information was disclosed by the authors:
Gerber Foundation, the Michigan Bloodspot Environmental Epidemiology Project (BLEEP).
Canadian Institute of Health Research.
Fonds de Recherche du Québec en Santé.
Canada Research Chairs (CRC) program, McGill University.
Canada Foundation for Innovation.

Natural Sciences and Engineering Research Council of Canada (NSERC) Discovery Grant.

## Competing Interests

The authors declare there are no competing interests.

## Author Contributions

- Jessica Pawly and Niladri Basu conceived and designed the experiments, performed the experiments, analyzed the data, contributed reagents/materials/analysis tools, prepared figures and/or tables, performed the computation work, authored or reviewed drafts of the paper, approved the final draft.
- Richard L. Neitzel conceived and designed the experiments, contributed reagents/materials/analysis tools, authored or reviewed drafts of the paper, approved the final draft.

## Ethics

The following information was supplied relating to ethical approvals (i.e., approving body and any reference numbers):

Institutional Review Board (IRB) approval for this work was obtained from McGill University (A06-M29-16B), the University of Michigan (HUM000771006), and the Michigan Department of Health and Human Services (201212-05-XA-R).

## Data Availability

The raw data is available in a Supplementary File.

## Supplemental Information

Supplemental information for this article can be found online at http://dx.doi.org/10.7717/peerj-achem.1#supplemental-information.

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
