# Peer review of "Analysis of copper, selenium, and zinc in newborn dried bloodspots using total reflection X-ray fluorescence (TXRF) spectroscopy"

_PeerJ Analytical Chemistry, doi:10.7717/peerj-achem.1_

## Round 0.1 · original submission · Minor Revisions

According to the reviewer reports, this work has merit. However, the reviewers suggest some changes and additions to improve the report. Please respond to their comments in an appropriate revision

·

Basic reporting

there is no language problem - perfectly written throughout
literature references adequate
abstract, text and the complete submitted article is well structured and clear

I am missing in the abstract and the paper the following information:
1. why are the chosen elements Cu Zn and Se of importance
2. give some examples on the importance of these elements on the nutrional status of new born and in general what more could be expected in understanding the composition of DBS
3. using TXRF you get multelement information in each spectrum were there other elements considered carrying interesting information
4. In conclusion mention some examples instead of "great demand across clinical and public health

Experimental design

Details in sample preparation would be fine
1. p7/111 back up the assumption that in a 3mm punch yoiu find 3,1 ┬Ál blood
2. p7/114 what parameters do you study to get a suitable analytical method
3. was the purity of HCl checked by TXRF
4. PVA 1% how much was added ?
5. p7/134 the reflectors were siliconized by SERVA solution to prepare the surface of the reflectors
6. p8/141 a photo of the final sample preparation on the reflector is informative pleae add
7. p8/142 one spectrum of the DBS measured should be added to show the spectroscopic details
8. p10/260 any idea why the Zn was lower in frozen cards??
9. p11/311 does the filter paper blank contain any of the targeted elements Cu Se Zn??
p12/343 Mg can not be measured in air, He flush or vacuum required

Validity of the findings

The use of TXRF for this application is a meaningful approach but includes the problem of dissolving the filter material Whatman 903 - nevertheless results are looking promising and fit in a general technique used in sampling minute volumes of blood which is advantageous.
Specifying more details on a medical fundamental research field where these information are important is required.

Additional comments

Some technical and spectroscopic information in particular sample preparation and influence of blank measurements on filter paper are missing. Specify if possible contamination or production impurities in the filter paper can lead to uncontrolled errors in the results. Point to the multelement capacity of TXRF so also cross readings among other elements are possible.

Reviewer 2 ·

Basic reporting

.

Experimental design

.

Validity of the findings

.

Additional comments

Very informative and interesting work. A useful application of TXRF for analysis of biological samples which are available in less quantity.

- Q1: Line 129 : The sample was then vortexed thoroughly and digested for 1.5 hours at 550C. At 550C the blood sample and the filter paper both get digested?

- Q2: In Line 133 : what was the purpose to add polyvinyl alcohol (1%) ?

- Q3: Why Cu, Se and Zn were studied? Do these elements have special significance? Please explain.

- Q4: please give some comments on the Variation in analytical results with respect to time and temp?

---

## Round 0.2 · accepted · Accept

The reviewer has recommended publication of the manuscript in the current form. Congratulations!

·

Basic reporting

The authors followed the sugegstions and for me the paper gained now successfully information for other readers

Experimental design

ok

Validity of the findings

ok

Additional comments

Interesting results and now informative in details